# Chitosan Oligosaccharide Inhibits the Synthesis of Milk Fat in Bovine Mammary Epithelial Cells through AMPK-Mediated Downstream Signaling Pathway

**DOI:** 10.3390/ani12131692

**Published:** 2022-06-30

**Authors:** Jing Fan, Jiayi Chen, Haochen Wu, Xin Lu, Xibi Fang, Fuquan Yin, Zhihui Zhao, Ping Jiang, Haibin Yu

**Affiliations:** 1College of Coastal Agricultural Sciences, Guangdong Ocean University, Zhanjiang 524088, China; fj13359819865@163.com (J.F.); chen17852729978@163.com (J.C.); 13275125309@163.com (H.W.); yinfuquan01@163.com (F.Y.); zhzhao@gdou.edu.cn (Z.Z.); 2The Key Laboratory of Animal Resources and Breed Innovation in Western Guangdong Province, Guangdong Ocean University, Zhanjiang 524088, China; 3College of Animal Science, Jilin University, Changchun 130062, China; luxin20@mails.jlu.edu.cn (X.L.); fangxibi@jlu.edu.cn (X.F.)

**Keywords:** bovine mammary epithelial cells, chitosan oligosaccharide, milk fat synthesis, AMPK signalling pathway

## Abstract

**Simple Summary:**

In order to study the effect of chitosan oligosaccharides on milk fat synthesis of bovine mammary epithelial cells (BMECs), we did a series of related experiments. The results showed that chitosan oligosaccharide (COS) could inhibit the fatty acid synthesis and promote milk fat decomposition and oxidation through *AMPK*/*SREBP1*/*SCD1*, *AMPK*/*HSL* and *AMPK*/*PPARα* signaling pathways to reduce the milk fat content in bovine mammary epithelial cells. We elucidated the important role of COS in BMECs lipid metabolism. COS may be the potential small-molecule component in milk cow molecular breeding to regulate milk fat synthesis and metabolism. These findings will help us to further understand the mechanism of COS on milk fat metabolism.

**Abstract:**

Chitosan oligosaccharide (COS) is a variety of oligosaccharides, and it is also the only abundant basic amino oligosaccharide in natural polysaccharides. Chitosan oligosaccharide is a low molecular weight product of chitosan after enzymatic degradation. It has many biological effects, such as lipid-lowering, antioxidant and immune regulation. Previous studies have shown that chitosan oligosaccharide has a certain effect on fat synthesis, but the effect of chitosan oligosaccharide on milk fat synthesis of bovine mammary epithelial cells (BMECs) has not been studied. Therefore, this study aimed to investigate chitosan oligosaccharide’s effect on milk fat synthesis in bovine mammary epithelial cells and explore the underlying mechanism. We treated bovine mammary epithelial cells with different concentrations of chitosan oligosaccharide (0, 100, 150, 200, 400 and 800 μg/mL) for 24 h, 36 h and 48 h respectively. To assess the effect of chitosan oligosaccharide on bovine mammary epithelial cells and determine the concentration and time for chitosan oligosaccharide treatment on cells, several in vitro cellular experiments, including on cell viability, cycle and proliferation were carried out. The results highlighted that chitosan oligosaccharide (100, 150 μg/mL) significantly promoted cell viability, cycle and proliferation, increased intracellular cholesterol content, and reduced intracellular triglyceride and non-esterified fatty acids content. Under the stimulation of chitosan oligosaccharide, the expression of genes downstream of Phosphorylated AMP-activated protein kinase (*P-AMPK*) and AMP-activated protein kinase (*AMPK*) signaling pathway changed, increasing the expression of peroxisome proliferator-activated receptor alpha (*PPARα*) and hormone-sensitive lipase (*HSL*), but the expression of sterol regulatory element-binding protein 1c (*SREBP1*) and its downstream target gene stearoyl-CoA desaturase (*SCD1*) decreased. In conclusion, these results suggest that chitosan oligosaccharide may inhibit milk fat synthesis in bovine mammary epithelial cells by activating the AMP-activated protein kinase signaling pathway, promoting the oxidative decomposition of fatty acids and inhibiting fatty acid synthesis.

## 1. Introduction

Milk is one of the primary sources of animal protein, amino acids, calcium and vitamins in the dietary structure. As a kind of natural fat, milk fat is an important part of milk, and it contains a variety of essential fatty acids and fat-soluble vitamins [1]. Therefore, milk fat has become one of the main indicators to measure milk quality. However, the incidence rate of diabetes, obesity and cardiovascular diseases has increased rapidly in recent years, leading to significant demand for low-fat or skim milk. Many studies have shown that milk fat synthesis is regulated by many factors, among which nutrition is one of the main factors. We hope to add certain nutrients to reduce the fat content in milk while making BMECs increase or have no toxic effect on BMECs to meet the needs of patients. A large number of studies have shown that COS can inhibit obesity and fat synthesis, but whether COS can regulate milk fat synthesis and its primary mechanism in BMECs has not been reported.

*AMPK* is an important serine/threonine protein kinase, which plays an important role as an energy “receptor” and “regulator” in cells [2]. In mammals, *AMPK* has seven different types of subunits, including α-Subunit (α1/α2), β-Subunit (β1/β2) and γ-Subunit (γ1/γ2/γ3) [3,4], where protein kinase AMP-activated catalytic subunit alpha 1 (AMPKα1) is mainly expressed in breast [5]. Phosphorylation α- subunit thr172 site is a necessary condition for the full activation of *AMPK*. Diabetes, obesity and lipid metabolism disorders were significantly associated with *AMPK* dysfunction, while *AMPK* agonist 5-aminoimidazole-4-carbox-amideribonucleoside (AICAR) could reduce serum lipid levels [6] in hyperlipidemia patients. Liang et al. [7] found that the liver fatty acid synthesis rate of mice with *SREBP-1c* gene knockout decreased by 50%, which shows that *SREBP-1c* plays a major role in regulating liver fatty acid synthesis. Studies have shown that *AMPK* inactivates *SREBP-1c* by directly phosphorylating the ser374 site [8], thereby down regulating the expression of *SREBP1c* and its target genes, including fatty acid synthase (*FASN*), acetyl CoA carboxylase (*ACACA*) and *SCD1* [9]. In addition, *PPARα* is essential for glucagon mediated fatty acid oxidation [10]. Bronner et al. [11] found in the experiment of culturing COS-7, HeLa and 293 cells in vitro that the increase of *AMPK* phosphorylation level after adding *AMPK* agonist AICAR can significantly promote the increase of *PPARα* protein expression, thus affecting fat oxidation. *HSL* is a crucial enzyme for intracellular triglyceride hydrolysis [12] and a determinant of fatty acid mobilization in fat and other tissues. In recent years, it has been found that liraglutide can increase the phosphorylation of *AMPK* and the expression levels of *ATGL* and *HSL* to promote the lipolysis pathway, reduce blood lipids and reduce abnormal lipid deposition in the kidney of diabetic nephropathy rats [13]. Therefore, *AMPK* may negatively regulate fat accumulation by inhibiting de novo fat synthesis and increasing oxidative hydrolysis of fatty acids.

As an important energy receptor and regulator in cells, *AMPK* can respond to hormone or nutritional signals in peripheral tissues to regulate the energy balance of the whole body. Studies showed that COS was added to normal and LPS-treated mouse macrophages. The phosphorylation level of AMPK protein in both mice was found to be increased. Through further experiments, the raw cells treated in advance by COS were stimulated with *AMPK* inhibitor DD and LPS. It was found that DD eliminated the effect of COS on down-regulating the proinflammatory cytokine gene expression level and up- regulating the mRNA expression level of the antioxidant enzyme gene [14]. Previous studies have shown that *AMPK* can regulate the expression of *SREBP-1c* and *PPARα*, which are important genes of fatty acid synthesis. Therefore, we believe COS may regulate cellular milk fat synthesis through the *AMPK* pathway. COS is mainly a natural, active substance extracted from the shells of marine shrimp and crabs. It is an oligosaccharide connected by D-glucosamine through the β-1, 4-glycosidic bond. Due to the advantages of low molecular weight, high degree of deacetylation, oligomerization and complete water solubility, COS can quickly enter cells and affect energy metabolism in cells. Previous studies have shown that COS alleviates metabolic disorders such as obesity [15], dyslipidemia [16], hyperglycemia and diabetes [17]. Choi et al. [18] Found that COS can alleviate the abnormal weight gain of mice caused by a high-fat diet (HFD) and reduce the lipid accumulation in serum and liver. Kang et al. [19] found that COS can inhibit pancreatic lipase activity and combine with bile acids to reduce intestinal fat absorption and increase fecal fat excretion.

Based on previous studies, we know that COS and *AMPK* signaling pathways can inhibit fat synthesis and accumulation, and COS can promote the phosphorylation level of *AMPK*. However, the role of COS in milk fat synthesis of BMECs and whether cos can regulate milk fat synthesis through the *AMPK* pathway are not clear. In this study, we verified the role of COS in BMECs by adding COS to BMECs; it was found that COS can pass through *AMPK*/*SREBP1*/*SCD1*, *AMPK*/*HSL* and *AMPK*/*PPARα* signal pathways, inhibits fatty acid synthesis and promotes milk fat decomposition and fatty acid oxidation, thus reducing the milk fat content in BMECs; it was clarified that cos might be a potential small-molecule component regulating milk fat synthesis and metabolism in dairy cattle molecular breeding, and COS could reduce the milk fat content in BMECs while not being toxic to BMECs. This result may contribute slightly to reducing the milk fat content to meet the needs of people with diabetes, obesity and cardiovascular diseases. The objectives of the present study were as follows: a) to explore the appropriate concentration and the optimal time for COS treatment on BMECs, and b) to explore the regulation mechanism of COS on milk fat metabolism through the AMPK signaling pathway in BMECs.

## 2. Materials and Methods

### 2.1. Drugs and Reagents

COS (average molecular weight ≈ 1000, degree of polymerization 3–7, purity 97%) are provided by the Yuan Ye Biology (Shanghai, China). DMEM and DPBS were purchased from HyClone (Logan, UT, USA), and fetal bovine serum (FBS) was purchased from Corning (Manassas, VA, USA). CCK8 kit was purchased from Zeta Life (Menlo Park, CA, USA), the Edu cell proliferation assay kit was purchased from RiboBio (Guangzhou, Guangdong, China), and Cell Cycle and Apoptosis Analysis Kit were purchased from Beyotime (Shanghai, China). The triglyceride and cholesterol detection kit was purchased from Pulilai (Beijing, China). The non-esterified fatty acids assay kit was purchased from Nanjing Jian cheng Bioengineering Institute (Nanjing, Jiangsu, China). The primary antibodies against *AMPKα1*, *SREBP1*, *SCD1* and *FASN* were purchased from Bioss (Beijing, China). The antibodies against *HSL* were purchased from Cell Signaling Technology (Boston, MA, USA). The antibodies against *β-actin* and Anti-rabbit secondary antibodies were purchased from Bioworld Technology CO., Ltd. (Minneapolis, MN, USA).

### 2.2. Cell Culture and Treatments

The BMECs used in the current study were isolated from a Chinese Holstein dairy cow’s mammary tissue and taken from the molecular genetics laboratory of the College of Animal Science, Jilin University [20,21]. Throughout the experiment, BMECs were cultured in an incubator at 37 °C with 5% carbon dioxide and grown in a basal DMEM/F12 medium containing 10% FBS. To determine the concentration and time for COS treatment of cells, when the cell confluence reached 70%, a total of 3 × 10^4^ isolated BMECs were inoculated into 96-well plates (Nest Biotechnology Co., Ltd., Shanghai, China); no more than 2.5 × 10^5^ BMECs were inoculated into the 24-well plates (Nest Biotechnology Co., Ltd., Shanghai, China), and an average of 1 × 10^6^ cells per well were inoculated into the 6-well plates (Nest Biotechnology Co., Ltd., Shanghai, China), respectively, and cells were treated with different concentrations of COS (0, 100, 150, 200, 400 and 800 μg/mL) for the time required for each experiment to complete the detection of CCK8, Edu and cell cycle progression. To detect the effect of COS on the milk fat metabolism of BMECs and its mechanism, when the cell confluence reached 70%, we spread the cells into 6-well plates, treated and collected the cells with the selected optimal culture concentration (100, 150 μg/mL) and time (24 h) to complete the triglycerides, cholesterol, free fatty acids, real-time quantitative PCR and Western blot tests.

### 2.3. CCK-8 and EdU (5-ethynyl-20-deoxyuridine) Assay

CCK8 assay was used to determine the effect of different concentrations of COS (0, 100, 150, 200, 400 and 800 μg/mL) on the viability of BMECs [22]. Cells were treated with different concentrations of COS (0, 100, 150, 200, 400 and 800 μg/mL) for 24, 36 and 48 h, respectively. Then, 10 μL of CCK8 solution was added and cultured at 37 °C for 2 h. The microplate reader detected the absorbance at 450 nm using a microplate reader (BioTek, VT, USA). Cell viability was expressed as the percentage of cells treated with COS (0 μg/mL). The cell proliferation was detected by EdU cell proliferation test kit [23]. The kit mainly includes three reagents: EdU solution, Apollo and Hoechst33.342. Edu is a thymine analogue, which can replace thymine in replicating DNA molecules during cell proliferation. Apollo and Hoechst33.342 are two fluorescent dyes. Apollo fluorescent dye and EdU can react specifically and emit red light to quickly detect cell DNA replication activity and reflect cell proliferation ability. Therefore, red fluorescence shows proliferation, and the cells emitting red fluorescence are positive. Hoechst33.342 fluorescent dye can specifically combine with DNA in the nucleus, showing the stained nucleus blue fluorescence under fixed excitation light and reflecting the number and density distribution of cells under the microscope. Edu was added to BMECs that had been treated with different concentrations of COS (0, 100, 150, 200, 400 and 800 μg/mL) for 24 h and co-cultured in an incubator for 3 h. After PBS cleaning, the cells were incubated with 4% paraformaldehyde at room temperature for 30 min to fix the cells. Then, for staining, Apollo dye solution and hoechst33342 dye solution were added and incubated in the dark for 30 min. The results were obtained through observation using a fluorescence microscope, and the number of positive staining cells was calculated by ImageJ software.

### 2.4. Detection of Cell Cycle Progression

The effects of different concentrations of COS (0, 100, 150, 200, 400 and 800 μg/mL) on the cell cycle of BMECs were detected by flow cytometry (Beckman Coulter, Brea, Florida, USA). The cells were treated with different concentrations of COS (0, 100, 150, 200, 400 and 800 μg/mL) for 24 h. The treated cells were collected and fixed with 70% ethanol at 4 °C for 24 h, then resuspended with a propidium iodide/RNase staining buffer (Cell Cycle and Apoptosis Analysis Kit) and incubated at 37 °C in the dark for 30 min. Finally, the proportion of G1 phase, S phase and G2/M phase cells was analyzed by flow cytometry and ModFit LT v3.1(Verity Software House, Topsham, ME, USA) software.

### 2.5. Determination of Triglyceride (TG) and Cholesterol Content

BMECs were treated with COS (100, 150 μg/mL) for 24 h. After the cells were lysed with lysate, the triglyceride and cholesterol concentrations in the cells were measured with a triglyceride kit and cholesterol kit. In addition, intracellular protein concentration was measured by a BCA protein assay kit (TaKaRa, Beijing, China). Finally, the triglyceride or cholesterol concentration/protein concentration ratio was used as the final data.

### 2.6. Detection of Non-Esterified Fatty Acids (NEFA) Content

The content of NEFAs in BMECs treated with COS (100, 150 μg/mL) for 24 h was detected by a non-esterified fatty acids assay kit. Briefly, after the cells were lysed with lysate, the cells were collected and centrifuged. A total of 4 μL of supernatant was placed in 96-well plates and mixed with 200 μL of a working solution and cultured at 37 ℃ for 5 min, and a microplate reader measured the absorbance as A1. Then, 50 μL of B working solution was added; the absorbance measured under the same conditions was A2. In addition, the protein concentration was detected using a BCA protein assay kit. Both absorbance values were measured at 546 nm, and the following formula calculated the intracellular NEFA content:


$$\begin{array}{c} \text{Intracellular NEFA concentration(mmol/gprot)}\\ \text{= \{(A2–A1) sample-(A2–A1) blank\}/\{(A2–A1) standard-(A2–A1) blank\} *}\\ \text{Standard concentration(mmol/L)/Intracellular protein}\\ \text{concentration(gprot/L)} \end{array}$$


### 2.7. Real-Time Quantitative PCR (RT-qPCR) and Western Blot Analysis

According to the steps of the RNA extraction kit (TaKaRa, Beijing, China), cells were lysed with buffer RL lysate containing 2% DTT to extract total RNA. The OD value and RNA concentration were evaluated using a nanodrop Lite (Thermo Scientific, Waltham, MA, USA) spectrophotometer. Then reverse transcription kit was used to transfer 1 μg RNA was reverse transcribed into cDNA. RT-qPCR primers (Table A1) were designed by primer5 software and analyzed by BLAST to confirm the specificity of each pair of primers to its target gene. The reaction system was 20 µL, including 5 µL of Chamq Universal SYBR qPCR master mix (Vazyme, Nanjing, China), 8.2 µL of double-distilled water, 0.4 µL of reverse and forward primers (per gene), and 1 µL of cDNA. Finally, a real-time PCR detection system (BIO-RAD, Hercules, CA, USA) was used to determine the gene expression, with *β-actin* used as an internal reference gene. The relative mRNA levels were calculated by the 2^−ΔΔCT^ method [24].

The medium was discarded, and the cells were washed with PBS twice. RIPA cell lysate (Meilunbio, Dalian, China) containing PMSF protease inhibitor (Beyotime, Shanghai, China) was added at the ratio of 100:1 to lyse cells. The cells were collected and centrifuged for 30 min, and then 10 μL of supernatant was diluted ten times, and the protein concentration was measured with a BCA protein assay kit. The protein was mixed with Tris-Tricinesds-SDS-PAGE loading buffer (Solarbio, Beijing, China) and RAPI lysate, and denatured at 95 °C for 5 min. Proteins were separated by SDS–polyacrylamide gel electrophoresis and then transferred to the activated PVDF membranes by wet electrophoretic transfer (Merck Millipore Ltd., Darmstadt, Germany). Subsequently, the membrane was closed in TBST solution (Sangon biotech, Shanghai, China) containing 5% skimmed milk powder (Coolaber, Beijing, China) at room temperature for 2 h, and then the membrane was cleaned. The washed PVDF membrane was incubated with primary antibody at 4 °C overnight. The next day, after cleaning the membrane, the PVDF membrane and secondary antibody were incubated at room temperature for 2 h. Finally, the membrane was cleaned again in the same way; protein bands were tested using the ECL chemiluminescence kit (BeyoECL Star, Shanghai, China), and analyzed by ImageJ software. The gray value of the target protein was divided by the gray value of *β-actin*, which was used as the final statistical value [25].

### 2.8. Statistical Analysis

All data are presented as means ± standard error from three independent experiments. Statistical analysis was performed using IBM SPSS Statistics 21 software (IBM, Armonk, New York, NY, USA). One-way analysis of variance was used to determine the differences between means. Statistical significance was declared at *p* < 0.05 or *p* < 0.01, which were represented by different superscript small letters and different superscript capital letters respectively.

## 3. Results

### 3.1. Effects of COS on BMECs Viability

The effects of COS on BMECs’ viability were detected using the CCK8 method to determine the optimal treatment time and concentration of the COS addition. The BMECs were treated with various concentrations of COS (100, 150, 200, 400 and 800 μg/mL) for 24 h, 36 h and 48 h, respectively. The results for CCK8 show that compared with the control group, when the BMECs were treated with an 800 μg/mL concentration of COS, whether for 24 h, 36 h or 48 h, the cell viability decreased significantly (*p* < 0.05); when the BMECs were treated with a 400 μg/mL concentration COS, the cell viability decreased significantly at 48 h (*p* < 0.05), and there was no significant difference between 24 h and 36 h. When the concentration of COS was 0–200 μg/mL, compared with the control group, the cell viability significantly improved when treated with 100 μg/mL and 150 μg/mL COS for 24 h, and the percentage of viable cells increased to 110.80% and 110.21%, respectively. There was no significant difference (*p >* 0.05) in cell activity when treated with 200 μg/mL COS for 24 h. After 36 h of treatment, the activity of cells treated with 100,150 and 200 μg/mL COS did not change significantly, and the percentage of viable cells was 107.82%, 106.87% and 105.74%, respectively. When the cells were treated with COS for 48 h, the percentage of viable cells increased to 106.67% (200 μg/mL). However, there was no significant difference (*p* > 0.05) in cell viability between cells treated with 100 μg/mL and 150 μg/mL concentrations of COS (Figure 1A). According to the test results for CCK8, COS had no toxic effect on BMECs in the range of 0-200 μg/mL. Additionally, for COS-treated BMECs, the percentage of viable cells was the highest at this concentration range at 24 h. Therefore, we chose 24 h as the best processing time.

### 3.2. Effects of COS on BMECs Proliferation

Next, we further explore the effect of COS on the proliferation of BMECs. Edu incorporation method was used to detect cell proliferation. The data showed that compared with the negative control group, lower COS concentration (100 and 150 μg/mL) obviously promoted mitotic activity of BMECs (*p* < 0.01), while high concentration COS (400 and 800 μg/mL) significantly reduced the mitotic activity of BMECs (*p* < 0.05) (Figure 1B,C). When the concentration of COS was 200 μg/mL, there was no significant effect on the mitotic activity of BMECs (Figure 1B,C). Based on the above data analysis, we preliminarily determined to treat the cells with 100 μg/mL and 150 μg/mL concentration of COS for 24 h to complete the follow-up experiment.

### 3.3. Effect of COS on Cell Cycle Progression

COS treatment (added to BMECs for 24 h) also affected the transformation of the BMECs cycle. The addition of COS promotes the process of the whole BMECs cycle, especially the progression of the cell cycle from S to G2. The flow cytometry assay demonstrated that as COS concentration increased, the proportion of cells in the G1 phase initially increased, whereas the proportion of cells in the S phase decreased gradually. The concentration of COS was 100 μg/mL and 150 μg/mL, increased the number of cells in the G2 phase and reached the peak. Whereas the concentration of COS was ≥ 200 μg/mL, the cell population in the G2 phase decreased with the enhance of concentration. When the concentration of COS was 800 μg/mL, the number of cells in the G2 phase reduced significantly (Figure 1D,E).

According to the detection results of CCK8, Edu incorporation and cell cycle, COS (100 and 150 μg/mL) were selected to treat BMECs for 24 h for the following experiments on the effect of COS on milk fat synthesis.

### 3.4. Effects of COS on BMECs Milk Fat Synthesis

To explore the changes in milk fat synthesis in BMECs after COS treatment, BMECs were incubated with COS (100 and 150 μg/mL) for 24 h, and the contents of TG, cholesterol and NFFA were measured. Our findings revealed that adding 100 μg/mL and 150 μg/mL concentrations of COS inhibited the synthesis of TG and NFFA. The BMECs TG and NFFA content was significantly lower in the COS (100 and 150 μg/mL) treatment group than in the control group (*p* < 0.05) (Figure 2A,B). On the contrary, when the concentration of COS was 100 μg/mL and 150 μg/mL, it could promote cholesterol synthesis, and the BMECs cholesterol contents were obviously higher in the COS (100 and 150 μg/mL) treatment group than in the control group (*p* < 0.01) (Figure 2C). In addition, COS (100 and 150 μg/mL) significantly decreased (*p* < 0.01) the mRNA level of TG and fatty acid synthesis related to diacylglycerol O-acyltransferase 1 (*DGAT1*), glycerol-3-phosphate acyltransferase (*GPAM*), *SCD1*, *FASN*, and *ACACA* genes (Figure 2E,G), whereas significantly increased (*p* < 0.05) TG and fatty acid decomposition related *HSL*, patatin like phospholipase domain containing 2 (*ATGL*), PPARG coactivator 1 beta (*PPARGC1B*), carnitine palmitoyltransferase 1A (*CPT1A*), and *PPARα* genes (Figure 2D,F). Thus, these data suggest that COS significantly inhibits the synthesis of TG and fatty acids in BMECs.

### 3.5. COS Inhibits Milk Fat Synthesis in BMECs through AMPK/SREBP1/SCD1, AMPK/HSL and AMPK/PPARα Signaling Pathways

After treating BMECs with COS, we found that some differentially expressed genes, such as *PPARα*, *HSL*, *ACACA*, *SCD1* and *FASN*, were related to the *AMPK* signaling pathway. Moreover, previous studies have shown that COS performs its biological functions by activating the *AMPK* signaling pathway. Compared with the control group, the expression of *AMPK* protein in BMECs treated with COS did not change significantly. Still, the transcription level of the *AMPK* gene, the protein expression of *P-AMPK* and the protein expression of *PPARα* and *HSL* downstream of *AMPK* increased significantly (*p* < 0.05) (Figure 3A,B,D,E,G). It was also found that compared with the control group, the COS treatment group had a significantly reduced (*p* < 0.01) *SREBP1* gene transcription level, 150 (μg/mL) concentration of COS significantly reduced (*p* < 0.01) *SREBP1* protein expression, and the 100 (μg/mL) concentration of COS tended to reduce its expression. Still, there was no significant difference (Figure 3A,C,F). In addition, RT-qPCR and Western blot results show that COS decreased the transcription level of target genes *SCD1*, *ACACA* and *FASN* downstream of *SREBP1* (Figure 2G) and the protein expression of the *SCD1* gene (Figure 3C,F). These results suggest that COS inhibits fatty acid synthesis and promotes fatty acid oxidation and triglyceride decomposition through *AMPK/SREBP1/SCD1*, *AMPK/PPARα* and *AMPK/HSL* signaling pathways in BMECs, and finally, inhibits milk fat synthesis (Figure 4).

## 4. Discussion

COS is a functional oligosaccharide produced by the deacetylation of chitin extracted from the shell of crustaceans such as shrimp and crabs. It is a natural oligosaccharide with a variety of beneficial activities. It can not only resist bacterial invasion [26], inflammation and tumors [27], but also has antioxidants [28] and antiobesity properties [15]. Because COS has a certain effect on lipid metabolism, COS was added to BMECs in this study to explore its effect on milk fat synthesis and its mechanism. The first thing we determined was the concentration of COS-treated cells. When the concentration of COS is low, it is preferentially located in mitochondria; however, when the concentration of COS is too high and the ability of mitochondria to bind COS is saturated, COS is not limited to mitochondria; it can be found in the cytoplasm and nucleus, and can be enriched in nuclear membrane and nucleolus [29]. It can be seen that different COS concentrations have different sub-localizations in cells, which suggests that the physiological function of COS may be related to the concentration. A high concentration of COS is not conducive to cell growth, and can promote tumor cell apoptosis. Huang et al. [30] pointed out that COS can cause DNA damage to cells. Previous studies pointed out that a high concentration of COS is located in the nucleolus, which also confirmed that cos can directly interact with DNA or protein in the nucleus, destroying cell function. This study used CCK8, cell cycle and cell proliferation experiments to determine the appropriate concentration of COS in mammary epithelial cells. We found that COS at concentrations of 100 and 150 (μg/mL) for 24 h not only presented no toxicity to cells, but also significantly promoted cell viability and cell proliferation. This treatment can also promote the process of the whole cell cycle, especially the transformation from the S phase to the G2 phase.

Similar to our study, one study shows that COS has a certain effect on adipocytes. COS can inhibit the differentiation of mouse adipocytes (3T3-L1) and reduce their triglyceride content and the expression of adipogenic marker genes [31]. In addition to regulating the fat metabolism of cells, cos has a good therapeutic effect on various diseases in clinical trials, such as hyperglycemia, hyperlipidemia, inflammation and tumors [32,33]. More importantly, some studies have emphasized the role of COS in regulating lipid metabolism in animal models [34,35]. Deng et al. [36], who built a diet-induced obesity model by feeding Sprague Dawley rats (obese rats) a high-fat diet and treated them with COS for 8 weeks, found that COS-treated rats (obese rats) had reduced weight gain, adipose tissue hypertrophy and proliferation were inhibited, and the fat weight ratio was reduced; COS improved dyslipidemia, reduced liver weight and organ index, inhibited liver lipid accumulation and prevented liver steatosis; in addition, it was also found that high-dose COS could increase the excretion of TGs in feces. After gavage of COS to Wistar rats fed a high-fat diet for three weeks, Wang et al. [37] found that COS had a beneficial regulatory effect on lipid abnormality induced by the HFD, which improved the plasma lipoprotein profile by reducing VLDL TGs, increasing HDL-C and decreasing apoB-containing particles; the potential of COS in anti-atherosclerosis is worth further study. In addition, Bai et al. [38] reported that COS could also improve liver glucose and lipid metabolism disorder by inhibiting the inflammatory response of HFD-fed mice. Tao et al. [14] reported that COS plays an important role in alleviating nonalcoholic fatty liver in mice.

In our study, the effect of COS on BMEC milk fat synthesis and its metabolic mechanism were analyzed for the first time, demonstrating the inhibitory effect of COS on milk fat synthesis in BMECs. Our results show that COS significantly reduced the content of triglycerides and free fatty acids in BMECs, and significantly reduced the mRNA expression of genes related to triglyceride and fatty acid synthesis in BMECs, such as *DGAT1*, *GPAM*, *SCD1*, *FASN* and *ACACA*. It also significantly increased the mRNA expression of related genes that promote the oxidative decomposition of triglycerides and free fatty acids in mammary epithelial cells, such as *HSL*, *ATGL*, *PPARα*, *PPARGC1B* and *CPT1A*. In addition, studies have found that COS can increase the concentration of plasma HDL-C (high-density lipoprotein cholesterol) in hyperlipidemic rats but has no significant effect on the concentration of LDL-C (low-density lipoprotein cholesterol) [37]. As we all know, HDL-C can transport cholesterol from peripheral tissues to the liver for reuse, prevent excessive accumulation of cholesterol in various tissues and improve lipid metabolism. It can be seen that COS may alleviate the disorder of lipid metabolism by regulating HDL-C. This study found that after adding 100 and 150 (μg/mL) COS, the total cholesterol content in BMECs increased significantly. However, total cholesterol content cannot reflect the content changes in other cholesterol types, such as HDL-C and LDL-C, in cells, and the main components of milk fat include 97–98% triglycerides, 0.2–0.4% cholesterol and 0.1% fatty acids [39]. The proportion of cholesterol is relatively small, so the increase in cholesterol does not affect the ability of chitosan oligosaccharides to inhibit milk fat synthesis.

Through the RT-qPCR experiment, we not only further verified the inhibitory effect of COS on milk fat, but also found that differentially expressed genes, such as *PPARα*, *HSL* and *SREBP1*, are downstream genes of the *AMPKα1* signaling pathway [40,41,42]. *SCD1*, *FASN* and *ACACA* genes are downstream target genes of the *SREBP1* gene [9]. Therefore, we suspect COS may inhibit milk fat synthesis through the *AMPK* signaling pathway. After the Western blot experiment, it was found that COS significantly promoted the expression and transcriptional activity of *PPARα* and *HSL* genes by activating *AMPK* and increasing the expression of *AMPK* to enhance the role of fatty acid oxidation and triglyceride decomposition in BMECs. On the other hand, activated *AMPK* inhibited the transcriptional activity and expression of *SREBP1* and down-regulated the transcriptional activity of its downstream target genes *SCD1*, *FASN* and *ACACA* to inhibit the synthesis of fatty acids in BMECs. The above biological processes eventually lead to the decrease in triglycerides and free fatty acids in BMECs.

In conclusion, our results show that treating BMECs with 100 and 150 (μg/mL) COS-concentrations for 24 h could promote the vitality, proliferation and cell cycle process of BMECs. COS inhibited the synthesis of milk fat in BMECs through *AMPK/SREBP1/SCD1*, *AMPK/PPARα* and *AMPK/HSL* signaling pathways. We elucidated the important role of COS in BMECs’ lipid metabolism. COS may be small molecules with potential for molecular enhancement of dairy cows to regulate the synthesis and metabolism of milk fats.

## Figures and Tables

**Figure 1 animals-12-01692-f001:**
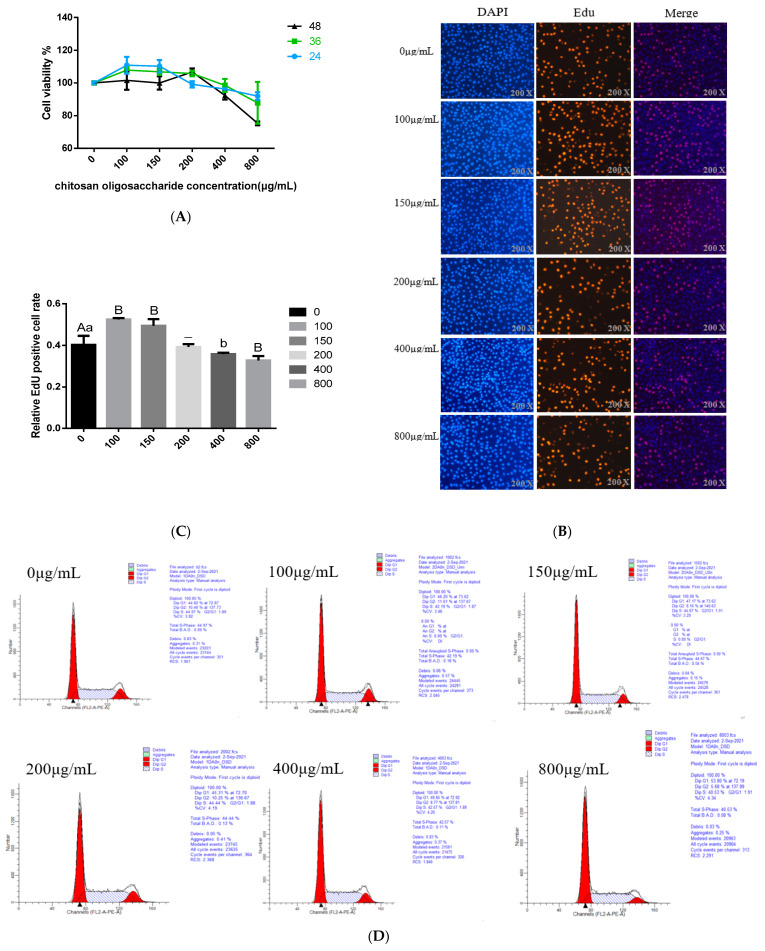
The effects of Chitosan oligosaccharide (COS) at different concentrations (0, 100, 150, 200, 400 and 800 μg/mL) on the viability, proliferation and cycle distribution of bovine mammary epithelial cells (BMECs) were observed after they were treated for 24, 36 and 48 h respectively. (**A**) CCK8 reagent was used to detect the BMECs viability at 450 nm. The *Y*-axis showed the percentage of cell survival, and the *X*-axis showed the doses of COS. (**B**) The proliferation of BMECs after COS treatment was measured by the Edu cell proliferation detection kit, the scale bar stands 200 μm. DAPI (blue), EdU (red). (**C**) the percentage of Edu positive cells in the total number of cells. (**D**) under COS culture at different concentrations, the BMECs cycle distribution was detected by flow cytometry. *Y*-axis represents the cell count analyzed, while *X*-axis indicates the DNA content of PI- stained cells. (**E**) The percentage of cells distributed in different stages of cell cycles (G1 phase, S phase and G2 phase) is represented by a histogram. Data were the mean ± SE from three independent experiments. Different small letters showed significant difference (*p* < 0.05), different capital letters showed extremely significant difference (*p* < 0.01), and the same letters or unmarked letters had no significant difference.

**Figure 2 animals-12-01692-f002:**
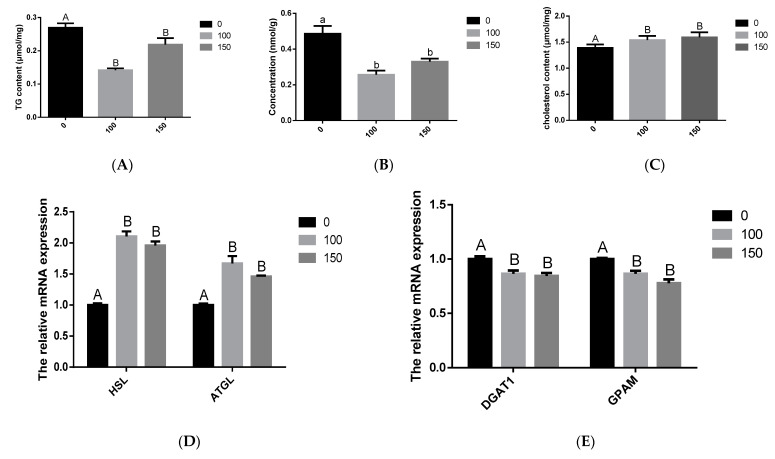
Effect of Chitosan oligosaccharide (COS) on milk fat synthesis of bovine mammary epithelial cells (BMECs). The levels of triglyceride (TG) (**A**), non-esterified fatty acids (NFFA) (**B**) and cholesterol (**C**) in BMECs treated with COS (0, 100, 150 μg/mL) for 24 h were measured. Further, verify that COS inhibits milk fat synthesis. The mRNA expression of *HSL* and patatin like phospholipase domain containing 2 (*ATGL*) related to triglyceride decomposition (**D**); the mRNA expression of genes diacylglycerol O-acyltransferase 1 (*DGAT1*) and glycerol-3-phosphate acyltransferase (*GPAM*) related to triglyceride synthesis (**E**); the mRNA expression of genes PPARG coactivator 1 beta (*PPARGC1B*), carnitine palmitoyltransferase 1A (*CPT1A*) and *PPARα* related to triglyceride decomposition (**F**); the mRNA expression of genes *SCD1*, *FASN* and acetyl-CoA carboxylase alpha (*ACACA*) related to free fatty acid synthesis (**G**). The data are presented as the mean ± SEM (*n* = 3). Different small letters showed significant difference (*p* < 0.05), different capital letters showed extremely significant difference (*p* < 0.01), and the same letters or unmarked letters had no significant difference.

**Figure 3 animals-12-01692-f003:**
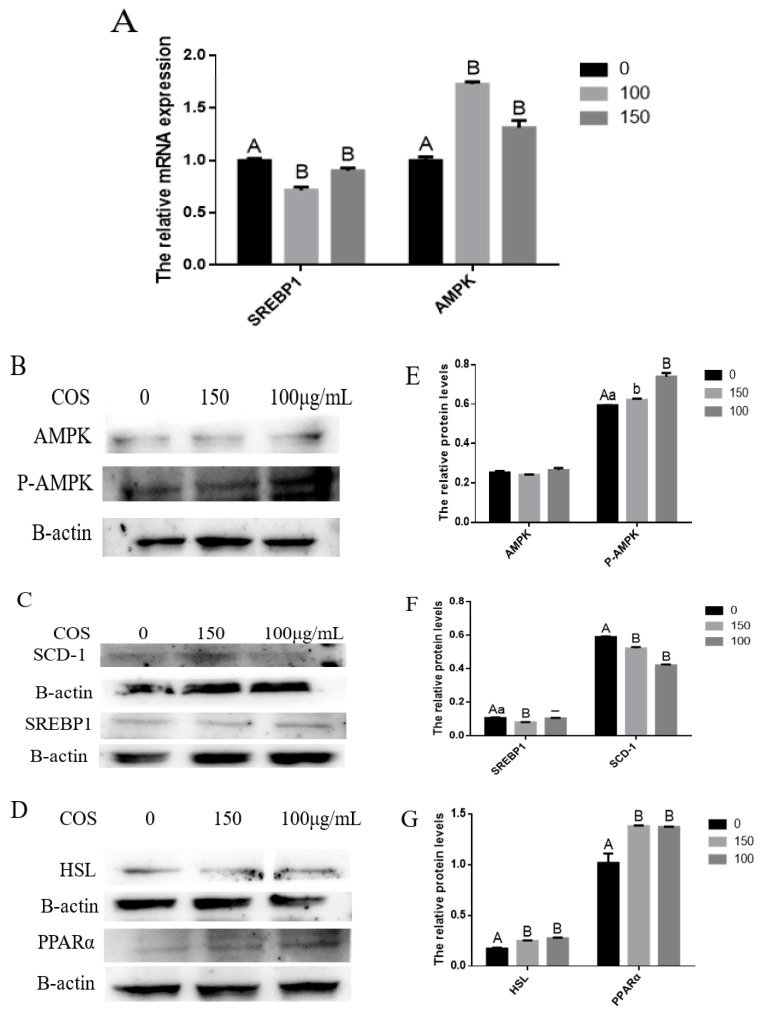
Chitosan oligosaccharide (COS) inhibits milk fat synthesis in bovine mammary epithelial cells (BMECs) via the AMP-activated protein kinase (*AMPK*) signaling pathway. BMECs were treated with different concentrations of COS (0, 100, 150 μg/mL) for 24 h. (**A**) Relative mRNA expression of *SREBP1* and *AMPK* in BMECs treated with COS. A greyscale scan quantified the relative folds of *AMPK* and its downstream genes protein levels from the western blots. *AMPK* and *p-AMPK* (**B**); *HSL* and *PPAR**α* (**C**); *SREBP1* and *SCD1* (**D**) protein levels were detected by Western blotting. Relative protein expression levels of *AMPK*, *p-AMPK*, *HSL*, *PPAR**α*, *SREBP1* and *SCD1* compared to that of *β-actin* (**E**–**G**). Data were the mean ± SE from three independent experiments. Different small letters showed significant difference (*p* < 0.05), different capital letters showed extremely significant difference (*p* < 0.01), and the same letters or unmarked letters had no significant difference.

**Figure 4 animals-12-01692-f004:**
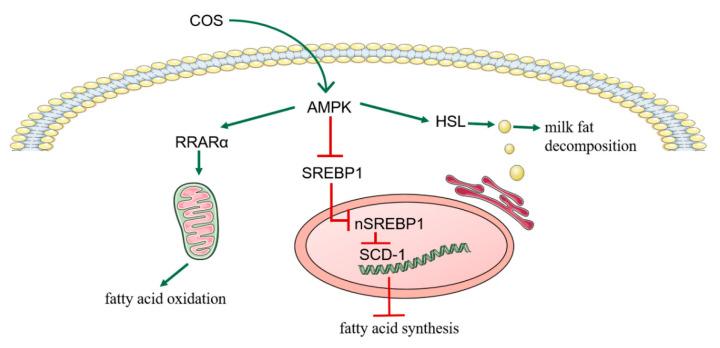
The above figure summarizes the mechanism of Chitosan oligosaccharide (COS) inhibiting milk fat synthesis in bovine mammary epithelial cells (BMECs). COS reduce the synthesis of milk fat in BMECs through *AMPK/SREBP1/SCD1*, *AMPK/PPARα* and *AMPK/HSL* signaling pathways.

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
