# Peer review of "Chitosan Oligosaccharide Inhibits the Synthesis of Milk Fat in Bovine Mammary Epithelial Cells through AMPK-Mediated Downstream Signaling Pathway"

_animals, 2022, doi:10.3390/ani12131692_

Round 1
Reviewer 1 Report
Introduction
Whilst the text is relevant to the study and very informative, the paper will benefit from transferring some of the information to the discussion to read in conjunction with the results of the present study.
Please describe the objective in a paragraph on its own.
Materials and methods
2.1. Were the authors fully OK with the antibodies against b-acting?
The other reagents are OK, we have used them as well without problems.
2.8. t-test is the wrong test for these results. First, please confirm normality of data and then reperform all the analysis with the correct test, according to normality. However, please do not use t-test, it is wrong.
The correct statistical analysis is important for proper assessment of the findings.
Results
Table 1. Please complete by including the annealing temperatures used. Then, transfer to supplementary material. Also, the information are related to M&M, not results.
The results need to be re-evaluated after performing correct analysis.
Discussion
Please include a new paragraph with the clinical consequences of the findings.
Overall. The manuscript needs re-assessment before possible acceptance.
Author Response
Dear Reviewer,
On behalf of all the authors of the submitted manuscript “Chitosan oligosaccharide inhibits the synthesis of milk fat in Bovine Mammary Epithelial Cells through AMPK mediated downstream signaling pathway” (ID: animals-1719689), we thank you so much for your kind reviews and meaningful suggestions. We have studied comments carefully and have made corrections which should meet with approvals we hope. The main revision in the paper (highlight changes in red) and the responses to the reviewers’ comments are as following:
Responds to Reviewer 1:
- Introduction. Whilst the text is relevant to the study and very informative, the paper will benefit from transferring some of the information to the discussion to read in conjunction with the results of the present study. Please describe the objective in a paragraph on its own.
Response: Thank you very much for your suggestion. Your suggestions have been very useful to us. We have described the objective separately in a paragraph, lines 114 to 125 in the manuscript. The contents are as follows: based on previous studies, we know that COS and AMPK signaling pathways can inhibit fat synthesis and accumulation, and COS can promote the phosphorylation level of AMPK. However, the role of COS in milk fat synthesis of BMECs and whether cos can regulate milk fat synthesis through the AMPK pathway is not clear. In this study, we verified the role of COS in BMECs by adding COS to BMECs; It is found that cos can pass AMPK/SREBP1/SCD1, AMPK/HSL, and AMPK/PPARα signal pathway inhibits fatty acid synthesis, promotes milk fat decomposition and fatty acid oxidation, thus reducing the milk fat content in BMECs; It was clarified that cos might be a potential small molecular component regulating milk fat synthesis and metabolism in dairy cattle molecular breeding, and COS could reduce the milk fat content in BMECs while not toxic to BMECs. This result may help to reduce the milk fat content in milk, thus providing genetic resources to meet the needs of patients with diabetes, obesity and cardiovascular disease for low milk fat dairy products.
- Materials and methods. (1) Were the authors fully OK with the antibodies against b-acting? The other reagents are OK, we have used them as well without problems.
Response: Thank you very much for your suggestion. There are many internal reference genes, such as: β-actin, GAPDH, Tub, 18s rRNA, TBP et al. β-actin was used to be the internal reference gene in this experiment, because, our previous study has shown that β-actin can be stably expressed in bovine mammary epithelial cells. Therefore, this experiment refers to the previous research and selects β- Actin was used as the internal reference gene [25] and was modified in line 236 of the manuscript.
(2) t-test is the wrong test for these results. First, please confirm normality of data and then reperform all the analysis with the correct test, according to normality. However, please do not use t-test, it is wrong. The correct statistical analysis is important for proper assessment of the findings.
Response: Thank you very much for your suggestion. The data obey normal distribution. We chose one-way ANOVA as the analysis method for all data, and modified the statistical analysis (see line 239), results, figures and notes (figure1, 2, 3) in the manuscript one by one. The revised results in the manuscript are respectively at line 249, line 258, line 288, line 318 and line 351.
- Results. Table 1. Please complete by including the annealing temperatures used. Then, transfer to supplementary material. Also, the information are related to M&M, not results. The results need to be re-evaluated after performing correct analysis.
Response: Thank you very much for your suggestion. We have filled in the annealing temperature in the table and transferred the table to the supplementary material (Appendix A see line 479).
- Discussion. Please include a new paragraph with the clinical consequences of the findings.
Response: Thank you very much for your suggestion. We have included an entire paragraph on the clinical consequences of the findings in the discussion section of the manuscript. The contents are as follows: similar to our study, the study shows that COS has a certain effect on adipocytes. COS can inhibit the differentiation of mouse adipocytes (3T3-L1) and reduce their triglyceride content and the expression of adipogenic marker genes [31]. In addition to regulating the fat metabolism of cells, cos has a good therapeutic effect on various diseases in clinical trials, such as hyperglycemia, hyperlipidemia, inflammation, and tumor [32, 33]. More importantly, some studies have emphasized the role of COS in regulating lipid metabolism in animal models [34, 35]. Deng et al [36] built a diet-induced obesity model by feeding Sprague-Dawley rats (obese rats) with a high-fat diet, and treating them with COS for 8 weeks , found that: COS-treated rats (obese rats) reduced weight gain, adipose tissue hypertrophy and proliferation were inhibited, and fat weight ratio was reduced; COS improved dyslipidemia, reduced liver weight and organ index, inhibited liver lipid accumulation, and prevented liver steatosis; In addition, it was also found that high-dose COS could increase the excretion of TG in faeces. After gavage of COS to Wistar rats fed with a high-fat diet for three weeks, Wang et al [37] found that COS had a beneficial regulatory effect on lipid abnormality induced by HFD, which improved plasma lipoprotein profile by reducing VLDL TG, increasing HDL-C and decreasing apoB-containing particles; The potential of COS in anti-atherosclerosis is worth further study. In addition, Bai et al [38] reported that COS could also improve liver glucose and lipid metabolism disorder by inhibiting the inflammatory response of HFD-fed mice. Tao et al [39] reported that COS plays an important role in alleviating nonalcoholic fatty liver in mice.

Reviewer 2 Report
We noticed many spelling errors in the manuscript
Line 37 and 145; it is written (0,100,150,200,400 37 and 800μg/mL) - should be written (0, 100, 150, 200, 400 37 and 800 μg/mL).
Line 42; it is written (100,150μg/mL) - it should be written (100, 150 μg/mL)
Line 77; it is written: Liang et al. found ... - should be written: Liang et al [7] found ...
Line 83; it is written: Bronnerli et al, ... - should be written: Bronner et al, [11] , ...
Line 109; it is written: Choi et al. Found that COS... - should be written: Choi et al [18] found that COS...
Line 111; it is written: Kang et al, found ... - should be written: Kang et al [19], found ...
Line 122; it is written: Purity 97%) ...- should be written: purity 97%) ...
Line 128; it is written: The Non-esterified fatty ...- should be written: The non-esterified fatty ...
Line 148; it is written: ... mechanism, When the cell ...- should be written: ... mechanism, when the cell ...
Line 162; it is written: Apollo and Hoechst 33.342 according to the instructions. - Clarify what the instructions are
Line 177 and 178; it is written: ... The ... - should be written: ... the ...
Line 206; it is written: ... method[23]. - should be written: ... method [23].
Line 247; it is written: ... to 106.6675% (200μg/mL). - Should be written: ... to 106.67% (200 μg/mL).
Figure 1. resolution is weak and not readable. Figure 1 needs to be corrected.
Line 375; it is written: Huang et al [28] pointed ...- should be written: Huang et al [28] pointed ...
Line 385; it is written: Choi et al, found that COS can inhibit the abnormal weight increase of mice fed with high-fat diet [29,30], and...- References [29,30] are incorrect (Jin et al. 2017; Kanneganti et al., 2012). All references should be checked.
Line 385; Authors' contributions: they should be written according to the instructions.
Line 443 to end of manuscript; references should be written according to instructions. Many errors were found in the references, in particular the references are unclear (Line 506; 26. B, D. N. N. A.; C, S. H. L.; D, M. M. K. and C, S. K. K. A. J. J. o. F. F. Production ...; Line 497; ... and Schedle, A. J. D. M. O. P. o. t. A. o. D. M. Initial ...; Line 489; ... Choi, K. C. J. L. A. R. 489 Modulation ...; Line 469; Line 512 and others)
Author Response
Dear Reviewer,
On behalf of all the authors of the submitted manuscript “Chitosan oligosaccharide inhibits the synthesis of milk fat in Bovine Mammary Epithelial Cells through AMPK mediated downstream signaling pathway” (ID: animals-1719689), we thank you so much for your kind reviews and meaningful suggestions. We have studied comments carefully and have made corrections which should meet with approvals we hope. The main revision in the paper (highlight changes in red) and the responses to the reviewers’ comments are as following:
Responds to Reviewer 2:
- We noticed many spelling errors in the manuscript
Response: Thank you very much for your suggestion. Your suggestions have been very useful to us. We have corrected the spelling mistakes you pointed out in the manuscript, and have asked a native English professional to correct the grammar and spelling mistakes in the manuscript. (See lines 36, 40, 75, 81, 107, 109, 126, 132, 151, 208, 219, 261).
- Line 162; it is written: Apollo and Hoechst 33.342 according to the instructions. - Clarify what the instructions are.
Response: Thank you very much for your suggestion. We have clarified the content of the instructions and made changes in line 163 of the manuscript. The contents are as follows: the cell proliferation was detected by EdU cell proliferation test kit [23]. The kit mainly includes three reagents: EdU solution, Apollo and Hoechst33.342. Edu is a thymine analogue, which can replace thymine into replicating DNA molecules during cell proliferation. Apollo and Hoechst33.342 are two fluorescent dyes. Apollo fluorescent dye and EdU can react specifically and emit red light, so as to quickly detect cell DNA replication activity and reflect cell proliferation ability. Therefore, red fluorescence shows proliferation, and the cells emitting red fluorescence are positive cells. Hoechst33.342 fluorescent dye can specifically combine with DNA in the nucleus, making the stained nucleus show blue fluorescence under fixed excitation light, and can reflect the number and density distribution of cells under the microscope. Edu was added to BMECs that had been treated with different concentrations of COS (0, 100, 150, 200, 400 and 800μg/mL) for 24 hours and co cultured in an incubator for 3 hours. After PBS cleaning, the cells were incubated with 4% paraformaldehyde at room temperature for 30 minutes to fix the cells. Then, Apollo dye solution and hoechst33342 dye solution were added in turn and incubated in dark for 30 minutes respectively for staining. The results were observed by fluorescence microscope, and the number of positive staining cells was calculated by ImageJ software.
- Figure 1. resolution is weak and not readable. Figure 1 needs to be corrected.
Response: Thank you very much for your suggestion. We have modified the resolution of figure 1 to 600 dpi.
- it is written: Choi et al, found that COS can inhibit the abnormal weight increase of mice fed with high-fat diet [29,30], and...- References [29,30] are incorrect (Jin et al. 2017; Kanneganti et al., 2012). All references should be checked.
Response: Thank you very much for your suggestion. We have deleted this error section and checked all references (see line 402).
- Authors' contributions: they should be written according to the instructions.
Response: Thank you very much for your suggestion. We have rewritten the author's contribution according to the instructions (see line 463).
- Line 443 to end of manuscript; references should be written according to instructions. Many errors were found in the references, in particular the references are unclear (Line 506; 26. B, D. N. N. A.; C, S. H. L.; D, M. M. K. and C, S. K. K. A. J. J. o. F. F. Production ...; Line 497; ... and Schedle, A. J. D. M. O. P. o. t. A. o. D. M. Initial ...; Line 489; ... Choi, K. C. J. L. A. R. 489 Modulation ...; Line 469; Line 512 and others).
Response: Thank you very much for your suggestion. We have rewritten and reviewed all references as instructed.

Reviewer 3 Report
The authors analyze the effect of Chitosan oligosaccharide on synthesis of milk fat in bovine mammary epithelial cells. The experiments and conclusions are appropriate, although some descriptions are hard to follow. I suggest an English revision and these replacements
Line 69 serine/threonine instead of silk/threonine
Lines 111 and 377 replace COS instead of cos
line245 reduce to only two decimal digits in the percentages
line 277 replace.
while higher COS concentration (200, 400, 800μg/mL) had no effect on mitotic activity (Figure 1B, C).
instead of
while higher COS concentration (200, 400, 800μg/mL) exhibited reduced mitotic activity of BMECs, but the difference was no statistical significance (Figure 1B, C).
line 427 replace.
COS may be small molecules with potential for molecular enhancement of dairy cows to regulate the synthesis and metabolism of milk fats.
Instead of
COS may be the potential small molecule components in milk cow molecular breeding to regulate milk fat synthesis and metabolism.
Author Response
Dear Reviewer,
On behalf of all the authors of the submitted manuscript “Chitosan oligosaccharide inhibits the synthesis of milk fat in Bovine Mammary Epithelial Cells through AMPK mediated downstream signaling pathway” (ID: animals-1719689), we thank you so much for your kind reviews and meaningful suggestions. We have studied comments carefully and have made corrections which should meet with approvals we hope. The main revision in the paper (highlight changes in red) and the responses to the reviewers’ comments are as following:
Responds to Reviewer 3:
- I suggest an English revision and these replacements Line 69 serine/threonine instead of silk/threonine.
Response: Thank you very much for your suggestion. Your suggestions have been very useful to us. We have corrected this error as you suggested (see line 67).
- Lines 111 and 377 replace COS instead of cos line245 reduce to only two decimal digits in the percentages.
Response: Thank you very much for your suggestion. Your suggestions have been very useful to us. We have corrected this error as you suggested (see line 109 and 390).
- line 277 replace.while higher COS concentration (200, 400, 800μg/mL) had no effect on mitotic activity (Figure 1B, C).instead of while higher COS concentration (200, 400, 800μg/mL) exhibited reduced mitotic activity of BMECs, but the difference was no statistical significance (Figure 1B, C).
Response: Thank you very much for your suggestion. Based on the opinions of other reviewers, we reanalyzed the data here and found that high concentrations of COS (400 and 800 μg/mL) significantly reduced the mitotic activity of BMEC (p<0.05). Therefore, the following modifications have been made: while high concentration COS (400 and 800 μg/mL) significantly reduced the mitotic activity of BMECs (P < 0.05) (Figure 1B, C). (see line 288).
- line 427 replace. COS may be small molecules with potential for molecular enhancement of dairy cows to regulate the synthesis and metabolism of milk fats Instead of COS may be the potential small molecule components in milk cow molecular breeding to regulate milk fat synthesis and metabolism.
Response: Thank you very much for your suggestion. We have corrected this error as you suggested (see line 460).

Reviewer 4 Report
Congratulations on this study. It appears sound, characterizing mechanisms of action of COS and highlighting them as a potentially useful supplement.
Overall the presentation is good, but I suggest that the text be passed through the free version of Grammarly or the authors avail themselves of the MPDI text editing services.
It seems quicker to recreate the document in MSword and edit that rather than write out line numbers. This is provided as a PDF which the authors can use copy and find to locate text. Comments are in italics. Suggestions for additions in red, and deletions struck through. Detailed edits were provided in the early parts of the document and later areas needing adjustment were highlighted.
Some of the text in some figures is unreadable so I suggest that the figures be separated so they can be enlarged.
Author Response
Dear Reviewer,
On behalf of all the authors of the submitted manuscript “Chitosan oligosaccharide inhibits the synthesis of milk fat in Bovine Mammary Epithelial Cells through AMPK mediated downstream signaling pathway” (ID: animals-1719689), we thank you so much for your kind reviews and meaningful suggestions. We have studied comments carefully and have made corrections which should meet with approvals we hope. The main revision in the paper (highlight changes in red) and the responses to the reviewers’ comments are as following:
Responds to Reviewer 4:
- Overall the presentation is good, but I suggest that the text be passed through the free version of Grammarly or the authors avail themselves of the MPDI text editing services.
Response: Thank you very much for your suggestion. Your suggestions have been very useful to us. We have asked a native English professional to correct the grammar and spelling mistakes in the manuscript and marked the revisions in red.
- Some of the text in some figures is unreadable so I suggest that the figures be separated so they can be enlarged.
Response: Thank you very much for your suggestion. We have modified the resolution of the figures in the manuscript to 600dpi and enlarged the figures.

Round 2
Reviewer 1 Report
The manuscript has been improved.
However, still, the objectives have not been clearly stated. The authors wrote a good summary of their hypothesis, but not objectives.
Please conclude the introduction with a sentence as follows:
The objectives of the present study are as follows: a) xxxx, b) yyyy and c) zzzz.
Then the manuscript can be accepted.
Author Response
Dear Reviewer,
On behalf of all the authors of the submitted manuscript “Chitosan oligosaccharide inhibits the synthesis of milk fat in Bovine Mammary Epithelial Cells through AMPK mediated downstream signaling pathway” (ID: animals-1719689), we thank you so much for your kind reviews and meaningful suggestions. We have studied comments carefully and have made corrections which should meet with approvals we hope. The main revision in the paper (highlight changes in red) and the responses to the reviewers’ comments are as following:
Responds to Reviewer 1:
- The manuscript has been improved. However, still, the objectives have not been clearly stated. The authors wrote a good summary of their hypothesis, but not objectives. Please conclude the introduction with a sentence as follows: The objectives of the present study are as follows: a) xxxx, b) yyyy and c) zzzz.
Response: Thank you very much for your suggestion. Your suggestions have been very useful to us. We concluded the introduction with the sentence you gave us. The contents are as follows: The objectives of the present study are as follows: a) to explore the appropriate concentration and the optimal time of COS acting on BMECs, b) to explore the regulation mechanism of COS on milk fat metabolism through AMPK signaling pathway in BMECs (See line 123).

Reviewer 2 Report
The proposed manuscript has been significantly improved.
a) The authors should carefully review the manuscript again and eliminate minor errors (line 69, line 81, ...).
b) Figures 1-3 should have a better resolution (the Figures are not readable).
c) I suggest that Appendix A becomes a separate .pdf file (if this Table A1 (Appendix A) remains in the manuscript, it should be after the References).
Author Response
Dear Reviewer,
On behalf of all the authors of the submitted manuscript “Chitosan oligosaccharide inhibits the synthesis of milk fat in Bovine Mammary Epithelial Cells through AMPK mediated downstream signaling pathway” (ID: animals-1719689), we thank you so much for your kind reviews and meaningful suggestions. We have studied comments carefully and have made corrections which should meet with approvals we hope. The main revision in the paper (highlight changes in red) and the responses to the reviewers’ comments are as following:
Responds to Reviewer 2:
- The proposed manuscript has been significantly improved. The authors should carefully review the manuscript again and eliminate minor errors (line 69, line 81, ...).
Response: Thank you very much for your suggestion. Your suggestions have been very useful to us. We have checked the manuscript and corrected the mistakes you pointed out and other minor mistakes. (See lines 69, 81).
- Figures 1-3 should have a better resolution (the Figures are not readable).
Response: Thank you very much for your suggestion. We have improved the resolution of figures 1-3 in the manuscript.
- I suggest that Appendix A becomes a separate .pdf file (if this Table A1 (Appendix A) remains in the manuscript, it should be after the References).
Response: Thank you very much for your suggestion. We have placed Table A1 in the manuscript after the references, and uploaded the PDF version of Table A1 in the supplementary document.
